# RESET-FREE LIFELONG LEARNING WITH SKILL-SPACE PLANNING

**Kevin Lu**
UC Berkeley
kzl@berkeley.edu

**Aditya Grover**
UC Berkeley
adityag@cs.stanford.edu

**Pieter Abbeel**
UC Berkeley
pabbeel@cs.berkeley.edu

**Igor Mordatch**
Google Brain
imordatch@google.com

## ABSTRACT

The objective of *lifelong* reinforcement learning (RL) is to optimize agents which can continuously adapt and interact in changing environments. However, current RL approaches fail drastically when environments are non-stationary and interactions are non-episodic. We propose *Lifelong Skill Planning* (LiSP), an algorithmic framework for non-episodic lifelong RL based on planning in an abstract space of higher-order skills. We learn the skills in an unsupervised manner using intrinsic rewards and plan over the learned skills using a learned dynamics model. Moreover, our framework permits skill discovery even from offline data, thereby reducing the need for excessive real-world interactions. We demonstrate empirically that LiSP successfully enables long-horizon planning and learns agents that can avoid catastrophic failures even in challenging non-stationary and non-episodic environments derived from gridworld and MuJoCo benchmarks[1].

## 1 INTRODUCTION

Intelligent agents, such as humans, continuously interact with the real world and make decisions to maximize their utility over the course of their lifetime. This is broadly the goal of lifelong reinforcement learning (RL), which seeks to automatically learn artificial agents that can mimic the continuous learning capabilities of real-world agents. This goal is challenging for current RL algorithms as real-world environments can be non-stationary, requiring the agents to continuously adapt to changing goals and dynamics in robust fashions. In contrast to much of prior work in lifelong RL, our focus is on developing RL algorithms that can operate in *non-episodic* or "reset-free" settings and learn from both online and offline interactions. This setup approximates real-world learning where we might have plentiful logs of offline data but resets to a fixed start distribution are not viable and our goals and environment change. Performing well in this setting is key for developing autonomous agents that can learn without laborious human supervision in non-stationary, high-stakes scenarios.

However, the performance of standard RL algorithms drops significantly in non-episodic settings. To illustrate this issue, we first pre-train agents to convergence in the episodic Hopper environment (Brockman et al., 2016) with state-of-the-art model-free and model-based RL algorithms: Soft Actor Critic (SAC) (Haarnoja et al., 2018) and Model-Based Policy Optimization (MBPO) (Janner et al., 2019), respectively. These agents are then trained further in a reset-free setting, representing a real-world scenario where agents seek to improve generalization via continuing to adapt at a test time where resets are more expensive. The learning curves are shown in Figure 1. In spite of near-perfect initialization, all agents proceed to fail catastrophically, suggesting that current gradient-based RL methods are inherently unstable in non-episodic settings.

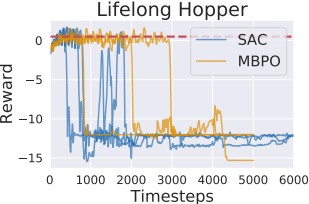

Figure 1: RL without planning fails without resets. Each line is one seed. The red line shows reward with no updates (i.e. frozen weights).

---

[1]Project website and materials: https://sites.google.com/berkeley.edu/reset-free-lifelong-learning

This illustrative experiment complements prior work highlighting other failures of RL algorithms in non-stationary and non-episodic environments: Co-Reyes et al. (2020) find current RL algorithms fail to learn in a simple gridworld environment without resets and Lu et al. (2019) find model-free RL algorithms struggle to learn and adapt to nonstationarity even with access to the ground truth dynamics model. We can attribute these failures to RL algorithms succumbing to *sink states*. Intuitively, these are states from which agents struggle to escape, have low rewards, and suggest a catastrophic halting of learning progress (Lyapunov, 1992). For example, an upright walking agent may fall over and fail to return to a standing position, possibly because of underactuated joints. A less obvious notion of sink state we use is that the agent simply fails to escape from it due to low learning signal, which is almost equally undesirable. A lifelong agent must seek to avoid such disabling sink states, especially in the absence of resets.

We introduce *Lifelong Skill Planning* (LiSP), an algorithmic framework for reset-free, lifelong RL that uses long-horizon, decision-time planning in an abstract space of *skills* to overcome the above challenges. LiSP employs a synergistic combination of model-free policy networks and model-based planning, wherein we use a policy to execute certain skills, planning directly in the skill space. This combination offers two benefits: (1) skills constrain the search space to aid the planner in finding solutions to long-horizon problems and (2) skills mitigate errors in the dynamics model by constraining the distribution of behaviors. We demonstrate that agents learned via LiSP can effectively plan for longer horizons than prior work, enabling better long-term reasoning and adaptation.

Another key component of the LiSP framework is the flexibility to learn skills from both online and offline interactions. For online learning, we extend Dynamics-Aware Discovery of Skills (DADS), an algorithm for unsupervised skill discovery (Sharma et al., 2019), with a *skill-practice* proposal distribution and a primitive dynamics model for generating rollouts for training. We demonstrate that the use of this proposal distribution significantly amplifies the signal for learning skills in reset-free settings. For offline learning from logged interactions, we employ a similar approach as above but with a modification of the reward function to correspond to the extent of disagreement amongst the models in a probabilistic ensemble (Kidambi et al., 2020).

Our key contributions can be summarized as follows:

- We identify skills as a key ingredient for overcoming the challenges to achieve effective lifelong RL in reset-free environments.

- We propose Lifelong Skill Planning (LiSP), an algorithmic framework for reset-free lifelong RL with two novel components: (a) a skill learning module that can learn from both online and offline interactions, and (b) a long-horizon, skill-space planning algorithm.

- We propose new challenging benchmarks for reset-free, lifelong RL by extending gridworld and MuJoCo OpenAI Gym benchmarks (Brockman et al., 2016). We demonstrate the effectiveness of LiSP over prior approaches on these benchmarks in a variety of non-stationary, multi-task settings, involving both online and offline interactions.

## 2 BACKGROUND

**Problem Setup.** We represent the lifelong environment as a sequence of Markov decision processes (MDPs). The lifelong MDP $\mathcal{M}$ is the concatenation of several MDPs $(\mathcal{M}_i, T_i)$, where $T_i$ denotes the length of time for which the dynamics of $\mathcal{M}_i$ are activated. Without loss of generality, we assume the sum of the $T_i$ (i.e., the total environment time) is greater than the agent's lifetime. The properties of the MDP $\mathcal{M}_i$ are defined by the tuple $(\mathcal{S}, \mathcal{A}, \mathcal{P}_i, r_i, \gamma)$, where $\mathcal{S}$ is the state space, $\mathcal{A}$ is the action space, $\mathcal{P}_i : \mathcal{S} \times \mathcal{A} \times \mathcal{S} \to \mathbb{R}$ are the transition dynamics, $r_i : \mathcal{S} \times \mathcal{A} \to \mathbb{R}$ is the reward function, and $\gamma \in [0, 1)$ is the discount factor. Consistent with prior work, we assume $r_i$ is always known to the agent specifying the task; it is also easy to learn for settings where it is not known. We use $\mathcal{P}$ and $r$ as shorthand to refer to the current $\mathcal{M}_i$ with respect to the agent.

The agent is denoted by a policy $\pi : \mathcal{S} \to \mathcal{A}$ and seeks to maximize its expected return starting from the current state $s_0$: $\arg\max_\pi \mathbb{E}_{s_{t+1} \sim \mathcal{P}, a_t \sim \pi}[\sum_{t=0}^{\infty} \gamma^t r(s_t, a_t)]$. The policy $\pi$ may be implemented as a parameterized function or an action-generating procedure. We expect the agent to optimize for the current $\mathcal{M}_i$, rather than trying to predict the future dynamics; e.g., a robot may be moved to an arbitrary new MDP and expected to perform well, without anticipating this change in advance.

**Skill Discovery.** Traditional single-task RL learns a single parametrized policy $\pi_\theta(\cdot|s)$. For an agent to succeed at multiple tasks, we can increase the flexibility of the agent by introducing a set of latent *skills* $z \in [-1, 1]^{dim(z)}$ and learning a skill conditional policy $\pi_\theta(\cdot|s, z)$. As in standard latent variable modeling, we assume a fixed, simple prior over the skills $p(z)$, e.g., uniform. The learning objective of the skill policy is to maximize some intrinsic notion of reward. We denote the intrinsic reward as $\tilde{r}(s, z, s')$ to distinguish it from the task-specific reward defined previously. One such intrinsic reward, proposed in DADS (Sharma et al., 2019), can be derived from a variational approximation to the mutual information between the skills and next states $I(s'; z|s)$ as:

$$\tilde{r}(s, z, s') = \log \frac{q_\nu(s'|s, z)}{\frac{1}{L} \sum_{i=1}^{L} q_\nu(s'|s, z_i)} \qquad \text{where } z_i \sim p(z). \tag{1}$$

Here $q_\nu(s'|s, z)$ is a tractable variational approximation for the intractable posterior $p(s'|s, z)$. Intuitively, this $\tilde{r}$ learns predictable (via the numerator) and distinguishable (via the denominator) skills. Due to the mutual information objective, DADS also learns skills with high *empowerment*, which is useful for constraining the space of options for planning; we discuss this in Appendix C.

**Model-Based Planning.** Whereas RL methods act in the environment according to a parameterized policy, model-based planning methods learn a dynamics model $f_\phi(s_{t+1}|s_t, a_t)$ to approximate $\mathcal{P}$ and use Model Predictive Control (MPC) to generate an action via search over the model (Nagabandi et al., 2018b; Chua et al., 2018). At every timestep, MPC selects the policy $\pi$ that maximizes the predicted $H$-horizon expected return from the current state $s_0$ for a specified reward function $r$:

$$\pi^{MPC} = \arg \max_\pi \mathbb{E}_{a_t \sim \pi, s_{t+1} \sim f_\phi} \left[ \sum_{t=0}^{H-1} \gamma^t r(s_t, a_t) \right]. \tag{2}$$

We use Model Path Predictive Integral (MPPI) (Williams et al., 2015) as our optimizer. MPPI is a gradient-free optimization method that: (1) samples policies according to Gaussian noise on the optimization parameters, (2) estimates the policy returns, and (3) reweighs policies according to a Boltzmann distribution on the predicted returns. For more details, see Nagabandi et al. (2020).

For all dynamics models used in this work, we use a probabilistic ensemble of $N$ models $\{f_{\phi_i}\}_{i=0}^{N-1}$, where each model predicts the mean and variance of the next state. For MPC planning, the returns are estimated via trajectory sampling (Chua et al., 2018), where each policy is evaluated on each individual model for the entire $H$-length rollout and the returns are averaged. For policy optimization, each transition is generated by sampling from a member of the ensemble uniformly at random.

## 3 LIFELONG SKILL-SPACE PLANNING

In this section, we present Lifelong Skill Planning (LiSP), our proposed approach for reset-free lifelong RL. We provide an outline of LiSP in Algorithm 1. The agent initially learns a dynamics model $f_\phi$ and a skill policy $\pi_\theta$ from any available offline data. Thereafter, the agent continuously updates the model and policy based on online interactions in the reset-free lifelong environment. The agent uses skill-space planning to act in the environment and avoid sink states. In particular, LiSP learns the following distributions as neural networks:

- A primitive dynamics model $f_\phi$ used for both planning and policy optimization
- A low-level skill policy $\pi_\theta$ trained from generated model rollouts on intrinsic rewards
- A discriminator $q_\nu$ for learning the intrinsic reward using Sharma et al. (2019)
- A skill-practice distribution $p_\psi$ to generate a curriculum for training skills

### 3.1 MODEL-BASED SKILL DISCOVERY

Our goal is to learn a skill-conditioned policy $\pi_\theta(a|s, z)$. In order to minimize interactions with the environment, we first learn a model $f_\phi$ to generate synthetic rollouts for policy training. Since there is no start state distribution, the initialization of these rollouts is an important design choice.

---

**Algorithm 1:** Lifelong Skill Planning (LiSP)

---

Initialize true replay buffer $\mathcal{D}$, generated replay buffer $\hat{\mathcal{D}}$, dynamics model ensemble
  $\{f_{\phi_i}\}_{i=0}^{N-1}$, policy $\pi_\theta$, discriminator $q_\nu$, and skill-practice distribution $p_\psi$
**if** performing offline pretraining **then**
  |   Learn dynamics model $f_\phi$ and train policy $\pi_\theta$ with `UpdatePolicy` until convergence
**while** agent is alive at current state $s$ **do**
  |   Update dynamics model to maximize the log probability of transitions of $\mathcal{D}$
  |   Update policy models with `UpdatePolicy` $(\mathcal{D}, \hat{\mathcal{D}}, f_\phi, \pi_\theta, q_\nu, p_\psi)$
  |   Execute action from `GetAction` $(s, f_\phi, \pi_\theta)$ and add environment transition to $\mathcal{D}$

---

We propose to learn a skill-practice distribution $p_\psi(z|s)$ to define which skills to use at a particular state. $p_\psi$ acts as a "teacher" for $\pi_\theta$, automatically generating a curriculum for skill learning. We include visualizations of the learned skills on 2D gridworld environments in Appendix E.2.

To actually learn the policy, we use the model to generate short rollouts, optimizing $\pi_\theta$ with SAC, similar to model-based policy learning works that find long rollouts to destabilize learning due to compounding model errors (Janner et al., 2019). To initialize the rollout, we sample a state from the replay buffer $\mathcal{D}$ and a skill to practice via $p_\psi$. The next state is predicted by the dynamics model $f_\phi$, where the model used to make the prediction is uniformly sampled from the ensemble. The transition is added to a generated replay buffer $\hat{D}$; gradients do not propagate through $f_\phi$. Given minibatches sampled from $\hat{D}$, both $\pi_\theta$ and $p_\psi$ are independently trained using SAC to optimize the intrinsic reward $\tilde{r}_{adjusted}$. Intuitively $p_\psi$ only selects skills which are most useful from the current state, instead of arbitrary skills. This is summarized in Algorithm 2.

---

**Algorithm 2:** Learning Latent Skills

---

**Hyperparameters:** number of rollouts $M$, disagreement threshold $\alpha_{thres}$
**Function** `UpdatePolicy` (replay buffer $\mathcal{D}$, generated replay buffer $\hat{\mathcal{D}}$, dynamics model $f_\phi$,
  policy $\pi_\theta$, discriminator $q_\nu$, skill-practice distribution $p_\psi$) **:**
  |   **for** $i = 1$ **to** $M$ **do**
  |   |   Sample $s_0^i$ uniformly from $\mathcal{D}$ and latent $z^i$ from skill-practice $p_\psi(\cdot|s_0^i)$
  |   |   Generate $s_1^i := f_\phi(\cdot|s_0^i, \pi_\theta(\cdot|s_0^i, z^i))$ and add transition $(s_0^i, a^i, z^i, s_1^i)$ to $\hat{\mathcal{D}}$
  |   Update discriminator $q_\nu$ on $\{s_0^i, a^i, z^i, s_1^i\}_{i=1}^M$ to maximize $\log q_\nu(s_1|s_0, z)$
  |   Calculate intrinsic rewards $\tilde{r}_{adjusted}$ for $\hat{\mathcal{D}}$ with $q_\nu, \alpha_{thres}$ using Equations 1 and 3
  |   Update $\pi_\theta, q_\nu, p_\psi$ using SAC with minibatches from $\hat{\mathcal{D}}$

---

### 3.1.1 OFFLINE SKILL LEARNING

A key problem in offline RL is avoiding value overestimation, typically tackled via constraining actions to the support of the dataset. We can use the same algorithm to learn skills offline with a simple adjustment to $\tilde{r}$ based on the model disagreement (Kidambi et al., 2020). For hyperparameters $\kappa, \alpha_{thres} \in \mathbb{R}^+$, we replace the intrinsic reward $\tilde{r}$ with $\tilde{r}_{adjusted}$, penalizing $\tilde{r}$ with $-\kappa$ if the model disagreement exceeds than $\alpha_{thres}$. This penalty encourages the policy to stay within the support of the dataset by optimizing against an MDP which underestimates the value function. We approximate the expected $\ell_2$ disagreement in the mean prediction of the next state, denoted $\mu_{\phi_i}$ for model $i$, with a sample. This penalty captures the epistemic uncertainty and is shown in Equation 3.

$$\tilde{r}_{adjusted} = \begin{cases} \tilde{r} & \text{dis}(s,a) = \mathbb{E}_{i \neq j}[\left\|\mu_{\phi_i}(s,a) - \mu_{\phi_j}(s,a)\right\|_2^2] \leq \alpha_{thres} \\ -\kappa & \text{dis}(s,a) > \alpha_{thres} \end{cases} \tag{3}$$

### 3.2 PLANNING IN SKILL SPACE FOR RESET-FREE ACTING

As described in Section 1, we argue that the failures of RL in the lifelong setting arise chiefly from the naive application of model-free RL. In particular, it is imperative that the model not only be used

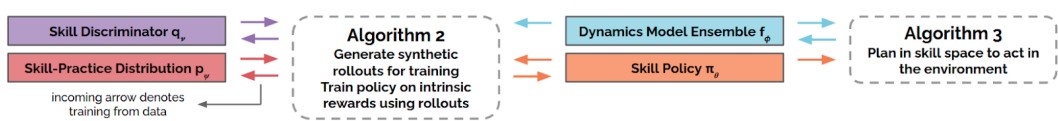

Figure 2: Schematic for Lifelong Skill Planning (LiSP). LiSP learns a set of skills using synthetic model rollouts and performs long-horizon planning in the skill-space for stable, safe lifelong acting.

for sample-efficient policy optimization (as in MBPO), but also that the model be used to safely act in the environment. The method in which acting is performed is important, serving both to exploit what the agent has learned and crucially to maintain the agent's safety in reset-free settings.

In this work, we propose to use model-based planning via MPC. We differ from prior MPC works in that we plans with the set of skills from Section 3.1, which allow us to utilize the broad capabilities of model-free RL while still enabling the benefits of model-based planning, namely exploration and long-horizon reasoning. The skills act as a prior for interesting options that the planner can utilize, seeking meaningful changes in the state. Also, the skills constrain the planner to a subset of the action space which the policy is confident in. The model-free policy learns to act in a reliable manner and is consequently more predictable and robust than naive actions. As a result, we are able to perform accurate planning for longer horizons than before (Chua et al., 2018; Nagabandi et al., 2020). Note that the complexity of our proposed approach is the same as prior MPC works, i.e. it is linear in the length of the horizon. We summarize this subroutine in Algorithm 3.

---

**Algorithm 3:** Skill-Space Planning

**Hyperparameters:** population size $S$, planning horizon $H$, planning iterations $P$, discount $\gamma$
**Function** `GetAction` (current state $s_0$, dynamics model $f_\phi$, policy $\pi_\theta$) :

> **for** $P$ planning iterations **do**
>
>> Sample skills $\{z^i\}_{i=1}^S \sim [-1,1]^{dim(z) \times H}$ based on distribution of previous iteration
>> Estimate returns $R = \{\sum_{t=0}^{H-1} \gamma^t r(s_t^i, \pi_\theta(\cdot|s_t^i, z_t^i), s_{t+1}^i)\}_{i=1}^S$ using trajectory sampling, with states $s^i$ sampled from $f_\phi$ for skills $z^i$
>> Use MPPI update rule on $R$ and $z$ to generate new distribution of skills $\{z_t\}_{t=0}^{H-1}$
>
> **return** $a \sim \pi_\theta(\cdot|s, z_0)$

---

We summarize the LiSP subroutines in Figure 2. Skills are first learned via Algorithm 2, wherein the skill discriminator generates the intrinsic rewards and the skill-practice distribution generates a skill curriculum. We then plan using the skill policy and the dynamics model as per Algorithm 3.

## 4 EXPERIMENTAL EVALUATIONS

We wish to investigate the following questions with regards to the design and performance of LiSP:

- Does LiSP learn effectively in lifelong benchmarks with sink states and nonstationarity?
- What are the advantages of long-horizon skill-space planning over action-space planning?
- Why is the skill-practice distribution important for learning in this setting?
- Does LiSP learn a suitable set of skills fully offline for future use?

**Lifelong learning benchmarks.** We extend several previously studied environments to the lifelong RL setting and distinguish these benchmarks with the prefix *Lifelong*. The key differences are that these environments have an online non-episodic phase where the agent must learn without resets and (optionally) an offline pretraining starting phase. The agent's replay buffer $\mathcal{D}$ is initialized with a starting dataset of transitions, allowing the agent to have meaningful behavior from the outset. More details are in Appendix E and we open source the implementations for wider use. Unless stated otherwise, we run 3 seeds for each experiment and our plots show the mean and standard deviation.

**Comparisons.** We compare against select state-of-the-art algorithms to help illustrate the value of different design choices. The closest baseline to ours is action-space MPC, also known as

PETS (Chua et al., 2018), which directly ablates for the benefit of using skills for planning. We also compare to SAC (Haarnoja et al., 2018), representing the performance of model-free RL on these tasks. Finally we consider MOReL (Kidambi et al., 2020), the offline model-based algorithm which proposed the disagreement penalty discussed in Section 3.1.1. Note that the latter two are principally single-task algorithms, while MPC is more suited for multiple reward functions. We believe these represent a diverse set of algorithms that could be considered for the reset-free setting.

### 4.1 EVALUATION ON LIFELONG BENCHMARKS

We begin by evaluating the overall LiSP framework (Algorithm 1) in non-episodic RL settings, particularly how LiSP interacts with sink states and its performance in nonstationary reset-free settings.

**Nonstationary Mujoco locomotion tasks.** We evaluate LiSP on Hopper and Ant tasks from Gym (Brockman et al., 2016); we call these Lifelong Hopper and Lifelong Ant. The agents seek to achieve a target forward x-velocity, which changes over the agent's lifetime. Learning curves are shown in Figure 3. Most of the LiSP seeds remain stable despite sink states and adapt instantly to the current tasks. As predicted by Figure 1, the SAC and MOReL agents are unstable and do poorly, fundamentally because their gradient updates are not stable. The long-horizon planning capability of LiSP is also crucial, enabling LiSP to outperform SAC and MOReL which lack the ability to plan. Furthermore, planning over the space of skills is necessary to achieve improvement over MPC, which is not capable of accurate planning on its own. LiSP outperforms all of the baselines tested.

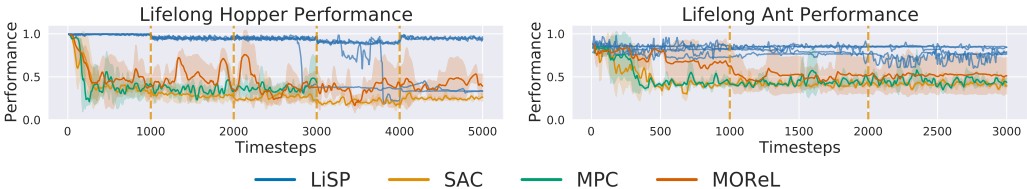

Figure 3: Learning without resets. Vertical lines denote task changes. Each blue line represents one seed of LiSP out of 5; the other algorithms have lower variance (since they fail), so we only show the mean of 3 seeds. Performance is normalized against 1 (for more details, see Appendix E.1.1).

**Minimizing resets with permanent sink states.** We evaluate each method in a lifelong 2D volcano gridworld environment where the agent navigates to reach goals while avoiding pitfalls which permanently trap the agent if there is no intervention. Every 100 timesteps, the pitfalls and goals rearrange, which allow the agent to get untrapped; we consider this a "reset" if the agent was stuck as it required this intervention. We use limited pretraining for this en-

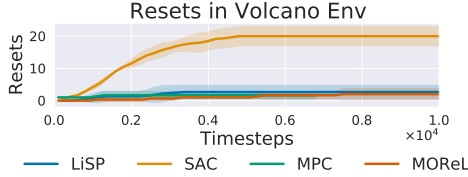

Figure 4: 2D Volcano environment.

vironment, i.e. we train the model but not the policies, reflecting the agent's ability to act safely while not fully trained. Figure 4 shows the number of times the agent got stuck during training. We find all the model-based methods, including LiSP, are easily capable of avoiding resets, suggesting model-based algorithms are naturally suited for these settings, incorporating data to avoid failures.

### 4.2 ADVANTAGES OF LONG HORIZON SKILL-SPACE PLANNING

Next, we seek to clearly show the advantages of planning in the skill space, demonstrating the benefits of using skills in Algorithm 3 and arguing for the use of skills more broadly in lifelong RL.

**Constraining model error.** We can interpret skill-space planning as constraining the MPC optimization to a more accurate subset of the action space which the policy operates in; this added accuracy is crucial for the improvement of LiSP vs MPC in the MuJoCo tasks. In Figure 5, we consider Lifelong Hopper and look at the one-step dynamics model error when evaluated on actions generated by the skill policy vs uniformly from the action space. Note this requires privileged access to the simulator so we cannot use these signals directly for planning. While most samples from both

have similar model error, the variance in the residual error for random actions is high, exhibiting a long tail. Even if the model is accurate for most samples, MPC can fatally overestimate on this tail.

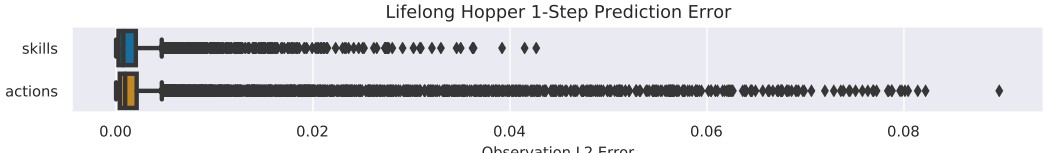

Figure 5: $\ell_2$ error in the next state prediction of the dynamics model $f_\phi$ for actions sampled from the skill policy vs. uniformly at random. The error variance is greatly reduced by only using skills.

**Constraining planning search space.** This advantage in the one-step model error extends to long-horizon planning. In the Lifelong Hopper environment, accurate long-horizon planning is critical in order to correctly execute a hopping motion, which has traditionally made MPC difficult in this environment. For good performance, we use a long horizon of 180 for planning. In Figure 6, we ablate LiSP by planning with actions instead of skills (i.e. MPC). Accurate planning with actions is completely infeasible due to the accumulated unconstrained model errors.

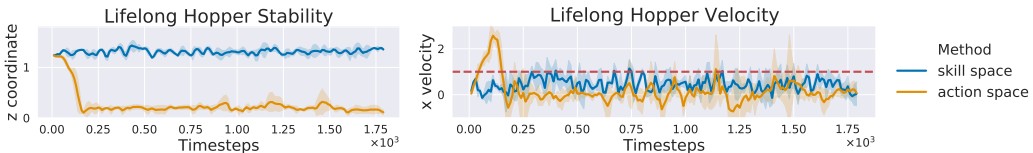

Figure 6: Long-horizon planning in Lifelong Hopper with skills instead of actions. Action-space planning is not stable with long horizons. The dashed red line denotes the target velocity.

**Model updates are extremely stable.** Furthermore, as described in Figure 1, we found that SAC and MBPO fail in reset-free settings when denied access to resets if they continue gradient updates, which we attributed to gradient-based instability. Here we try to understand the stability of RL algorithms. In particular, since we found that policy/critic updates are unstable, we can instead consider how model updates affect planning. To do so, we consider SAC as before, as well as action-space MPC over a short 5-step horizon trying to optimize the same SAC critic (tied to the policy); we note this is similar to Sikchi et al. (2020). The results are shown in Figure 7. If only the model is updated, planning is very stable, showing the resilience of planning to model updates; this best resembles LiSP, which does not rely on a policy critic for planning. Alternatively, value function updates make planning unstable, supporting the idea that long-horizon planning improves stability vs relying on a value function. However, even the short 5-step planner avoids catastrophic failures, as the additional model stability quickly neutralizes value instability.

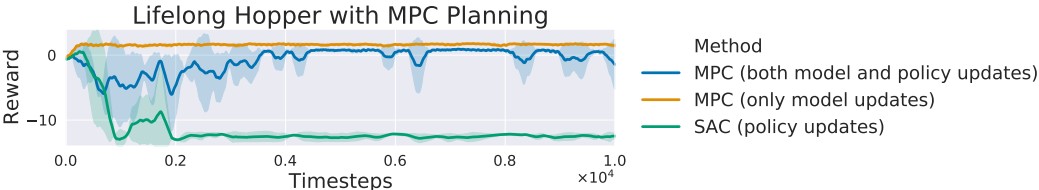

Figure 7: Pretrained action-space MPC agent with a terminal value function placed in a reset-free setting with continued gradient updates to the model and policy/value. Unlike the naive SAC agent that acts according to greedy "1-step" planning, the MPC agent plans vs the same critic over a short 5-step horizon and mostly avoids failure. Note this graph shows the same setting as Figure 1.

### 4.3 ONLINE SKILL LEARNING WITH A MODEL

In this section, we show the importance of the skill-practice distribution from Algorithm 2 for learning setups that require hierarchical skills and involve short model-based rollouts. The skill-practice distribution is crucial for improving learning signal, generating useful skills for planning.

**Hierarchical skill learning.** We consider a 2D Minecraft environment where the agent must learn hierarchical skills to build tools, which in turn are used to create higher level tools. We ablate against minimizing $\tilde{r}_{adjusted}$ and a baseline of not using a practice distribution; the former acts as a sanity check whereas the latter represents the previous convention in skill discovery literature. Note that we do not compare against MOReL here since the asymptotic performance will be similar to SAC, as they both rely on Q-learning policy updates. Similar to the volcano environment, we only do limited pretraining on a small set of starting data. The learning curves are in Figure 8. Learning skills with directed practice sampling is necessary for learning useful skills. Without the skill-practice curriculum, the asymptotic performance of LiSP is significantly limited. Additionally, the fact that MPC performs significantly worse than LiSP suggests that planning in the skill space – namely, planning in a space of expressive skills – greatly aids the search process and exploration.

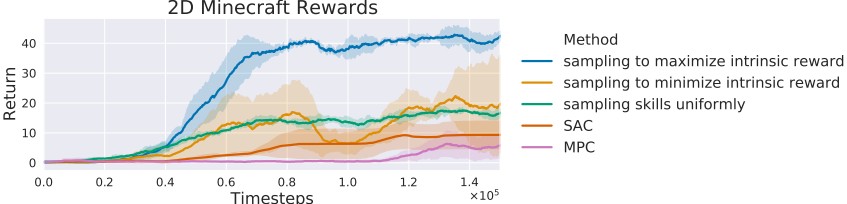

Figure 8: 2D Minecraft learning curves with varying skill-practice distributions. Practicing skills that maximize intrinsic reward outperforms naive methods for choosing which skills to practice.

**Learning with short rollouts.** Algorithm 2 learns skills using one-step rollouts from arbitrary states. These short rollouts significantly reduce learning signal relative to sampling a skill from a fixed start state distribution and running the skill for an entire episode: rather than being able to "associate" all the states along an episode with a skill, the agent will be forced to associate all states with all skills. Intuitively, the skill-practice distribution can alleviate this, associating states with certain skills which are important, making the skill learning process easier. We investigate this by analyzing DADS-Off (Sharma et al., 2020) – an off-policy improved version of DADS – and resampling skills every $K$ timesteps. This simulates $K$-step rollouts from arbitrary starting states, i.e. an oracle version of LiSP with data generated from the real environment, without a skill-practice distribution. We show the learned skills with various $K$ for Lifelong Ant in Figure 9. Despite DADS-Off using off-policy data, resampling skills hurts performance, leading to a loss of diversity in skills. When generating transitions with a model as in LiSP, it is unstable to run long model rollouts due to compounding errors, making the skill-practice distribution critical.

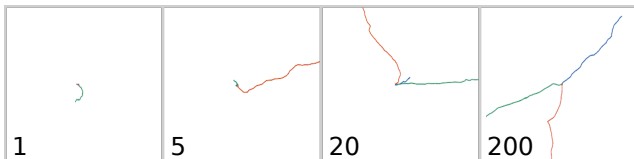

Figure 9: Skill learning collapses for DADS-Off in Ant, even without the primitive dynamics model $f_\phi$. Shown are the x-y positions, where each color denotes one skill starting from the origin. More diverse skills are better. From left to right, skills are resampled during training every 1 (fully online, like LiSP), 5, 20, and 200 (fully episodic) timesteps.

### 4.4 LEARNING SKILLS ENTIRELY FROM OFFLINE DATA

Finally, we evaluate the entire LiSP framework (Algorithm 1) in a fully offline setting without test-time updates, considering LiSP solely as an offline RL algorithm. Again, the most direct baseline/ablation to LiSP is action-space MPC, also known as PETS (Chua et al., 2018), which can learn

offline for multiple test-time tasks. We consider two versions: one that uses a long horizon for planning ($H = 180$, MPC-Long) and one with a short horizon ($H = 25$, MPC-Short). The former is a direct ablation, while the latter is closer to prior work and the PETS paper.

**Learning skills offline for multiple tasks.** We consider three Lifelong Hopper tasks, where tasks define target velocities. We use the same dataset as Section 4.1, collected from a forward hop task. Our results are in Table 1. LiSP successfully learns a set of skills offline for planning for all tasks.

Table 1: Average performance when learning fully offline from datasets for Lifelong Hopper. Though not explicitly designed for this setting, LiSP can learn skills offline for multiple tasks.

| Task | LiSP (Ours) | MPC-Long | MPC-Short |
|---|---|---|---|
| Forward Hop – Fast | **0.84 ± 0.02** | 0.27 ± 0.10 | 0.27 ± 0.04 |
| Forward Hop – Slow | **0.88 ± 0.05** | 0.49 ± 0.42 | 0.32 ± 0.08 |
| Backward Hop – Slow | **0.81 ± 0.01** | 0.38 ± 0.16 | 0.27 ± 0.06 |

## 5 DISCUSSION AND RELATED WORK

In this section, we provide an overview of other works related to LiSP, in similar fields to lifelong RL. An in-depth discussion of the related works is in Appendix A.

**Non-Episodic RL.** The traditional approach to non-episodic RL is to explicitly learn a policy to reset the environment (Even-Dar et al., 2005; Han et al., 2015; Eysenbach et al., 2017). This requires somewhat hard-to-define notions of what a reset should accomplish and still uses manual resets. We do not learn a reset policy, instead focusing on naturally safe acting via learning from offline data and effective planning. Zhu et al. (2020) and Co-Reyes et al. (2020) propose solutions to lack of learning signal in reset-free settings but only consider environments without sink states; the latter requires control over the environment to form a training curriculum. Lu et al. (2019) and Lowrey et al. (2018) highlight the benefits of both planning and model-free RL for lifelong agents but require highly accurate world models. Offline RL learns policies exclusively from offline data, which naturally lacks resets (Levine et al., 2020; Agarwal et al., 2020; Wu et al., 2019; Yu et al., 2020; Kumar et al., 2020), but most work is restricted to single-task, stationary settings.

**Model-Based Planning.** Existing works in model-based planning are restricted to short horizons (Chua et al., 2018; Wang & Ba, 2019; Nagabandi et al., 2020). Similarly to LiSP, some works (Kahn et al., 2017; Henaff et al., 2019) try to reduce model error for planning by penalizing deviations outside the training set. We found embedding uncertainty into the cost function causes poor cost shaping; these works also still have relatively short horizons. Mishra et al. (2017) learns action priors used to generate actions for MPC that embed past actions in a latent space for constrained planning, similarly to how we skill constraint, but only considers a fairly simple manipulation task. Some works (Lowrey et al., 2018; Lu et al., 2019; Sikchi et al., 2020), including in offline RL (Argenson & Dulac-Arnold, 2020), use terminal value functions to aid planning, allowing for successful short-horizon planning; as discussed previously, this does not directly translate to good performance in the lifelong setting due to instability and nontriviality in learning this function.

**Skill Discovery.** The key difference from prior skill discovery work is the lack of episodes. The most relevant work to ours is DADS (Sharma et al., 2019). Unlike DADS, we plan over the model with action predictions, which allows for accurate planning even if the discriminator is not accurate. Most other works (Achiam et al., 2018; Warde-Farley et al., 2018; Hansen et al., 2019; Campos et al., 2020) are complementary methods that can help learn a set of skills. Some seek to use skills as pretraining for episodic learning, as opposed to our focus on safe reset-free acting.

## 6 CONCLUSION

We presented LiSP, an algorithm for lifelong learning based on skill discovery and long-horizon skill-space planning. To encourage future work in this space, we proposed new benchmarks that capture the key difficulties of lifelong RL in reset-free, nonstationary settings. Our experiments showed that LiSP effectively learns in non-episodic settings with sink states, vastly improving over prior RL methods, and analyzed ablations to show the benefits of each proposed design choice.

### ACKNOWLEDGEMENTS

We would like to thank Archit Sharma for advice on implementing DADS.

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

## A    FURTHER DISCUSSION AND RELATED WORK

**Model-Based Planning.**    Chua et al. (2018) propose to use an ensemble of probabilistic dynamics models for planning, which we use, but by itself struggles to scale to higher-dimensional Mujoco tasks due to limited model accuracy; the use of model ensembling to mitigate errors has also been previously explored by Nagabandi et al. (2018b) and Kurutach et al. (2018). Wang & Ba (2019) improves upon some of the weaknesses of Chua et al. (2018) by planning in the parameter space of policies, solving some MuJoCo tasks asymptotically, but is still not accurate enough to plan over horizons long enough for environments that require a long sequence of coordinated actions. Nagabandi et al. (2020) shows that an improved optimizer can improve MPC performance for short-horizon tasks. We note that all of these methods struggle to plan for long horizons due to compounding model errors, and none have been able to perform MPC in the Hopper environment without a terminal value function (as in Sikchi et al. (2020) and Argenson & Dulac-Arnold (2020)), which is a novelty of our work.

Wang & Ba (2019), Lu et al. (2019), Argenson & Dulac-Arnold (2020), and Sikchi et al. (2020) initialize the actions for MPC with a prior policy, but this is not the same as constraining the space. Consequently, they do not have as strong long-horizon benefits, instead using other methods for attaining strong performance. Sometimes early termination and low Gaussian noise for random shooting is used to attempt to approximate a constraint (Sikchi et al., 2020), but this is not very effective in higher dimensional environments. Some works (Venkatraman et al., 2015; Mishra et al., 2017; Hafner et al., 2019) try to improve planning accuracy with multi-step prediction losses, which is complementary to our work and could improve performance. Furthermore, even though value functions for short horizon planning is not particularly promising, we note that terminal value functions still have other benefits that can improve performance when combined with long-horizon planning, such as improved exploration or long-term reasoning (Lowrey et al., 2018; Lu et al., 2019).

**Hierarchical RL.**    Our work is somewhat similar to hierarchical option-critic architectures that "plan" without MPC (Sutton et al., 1999; Bacon et al., 2017; Nachum et al., 2018; Khetarpal et al., 2020), which is sometimes referred to as "background-time planning" (Sutton & Barto, 2018). However, our skills are learned with an unsupervised reward and composed via MPC, which has the promise of more explicit long-term hierarchical benefits, whereas there is some evidence (Nachum et al., 2019) that policy-based hierarchy only aids in exploration – not long-term reasoning – and is replaceable with monolithic exploration methods. In contrast planning adds more explicit and interpretable "decision-time planning", which may have better scaling properties in more complex environments, as long as planning can be done accurately.

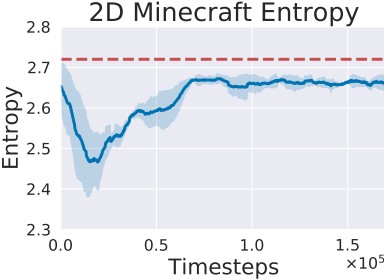

Figure 10: Entropy of skills proposed by skill-practice distribution. The red line shows the maximum possible entropy, which exists as the model class is a TanhGaussian (Haarnoja et al., 2018).

Our skill-practice distribution somewhat resembles the interest function of Khetarpal et al. (2020); both are designed to associate certain skills with certain states, forming a type of curriculum. In particular, they note that using a small number of skills improves learning early but can be asymptotically limiting, which has some similarity to curriculums generated from the skill-practice distribution. We show the entropy of the skills generated for the 2D Minecraft environment in Figure 10; the skill-practice distribution automatically generates a curriculum where it purposefully practices less skills (low entropy) early in training, and more skills (high entropy) later. Unlike Khetarpal et al. (2020), our skill-practice distribution is not part of the policy update and is only used to generate

a curriculum. We also note that while this means that transitions generated from the model are not drawn from the prior distribution in the objective, since the critic $Q(s, z, a)$ learns a target for the next state using the same $z$ as $Q(s', z, \pi(a|s', z))$, it is correct without importance sampling.

**Safety.** Zhang et al. (2020) perform risk-averse planning to embed safety constraints, but require a handcrafted safety function. This is similar to arguments from the Safety Gym work (Ray et al., 2019), which argues that the most meaningful way to formulate safe RL is with a constrained optimization problem. However, the safety in our work deals more with stability and control, rather than avoiding a cost function, so our notion of safety is generally different from these ideas. More concretely, the type of uncertainty and risk that are crucial to our setting are more epistemic, rather than aleatoric, so these types of works and distributional RL are not as promising. Consequently, we found that long-horizon planning and trying to perform accurate value estimation at greater scale to be more effective than constraining against a cost function or using explicit risk aversion. We note that some "empowerment"-style methods (Gregor et al., 2016; Karl et al., 2017; Eysenbach et al., 2018; Sharma et al., 2019) optimize an objective that aims to resemble a notion of stability; however, this metric is hard to estimate accurately in general, so we do not explicitly use it for any form of constrained planning. Zhao et al. (2021) learn a more accurate estimator of the empowerment, which could be useful for future work in safety.

**Nonstationarity.** Much work in nonstationarity focuses on catastrophic forgetting (Rusu et al., 2016; Kirkpatrick et al., 2017; Schwarz et al., 2018), which is not a primary goal of our work as we are primarily concerned with acting competently in the current MDP without explicitly trying to remember all previous or any arbitrary MDPs. Our approach to nonstationary lifelong learning follows Lu et al. (2019), but we do not require access to the ground truth dynamics $\mathcal{P}$ at world changes. Lecarpentier & Rachelson (2019) tackles nonstationarity in zero-shot for simple discrete MDPs by using risk-averse planning; our experiments generally suggested that accurate and scalable planning was more important than explicit risk aversion for our settings. Other works (Nagabandi et al., 2018a; Finn et al., 2019; Rolnick et al., 2018) are typically nonstationary at the abstraction of episodes, which generally does not require competencies such as online safe adaptation, and instead try to adapt quickly, minimize regret, or avoid catastrophic forgetting as previously discussed. Xie et al. (2020) is closer to our setting, as they consider nonstationarity within an episode, but they still require manual resets except in a 2D environment which does not contain sink states.

## B  Further Plots for Learning Dynamics

In this section, we provide additional plots, seeking to give more insights on the learning of LiSP from the MuJoCo experiments from Section 4.1.

In particular, we can consider the intrinsic reward, which is a proxy for the diversity of skills. Since the intrinsic reward is calculated under the model, which changes and has inaccuracies, high intrinsic reward under the model is not always the best indicator, whereas it is a more reliable metric when learned from real world transitions as in Sharma et al. (2020). Also, the intrinsic reward is sampled according to an expectation given by the skill-practice distribution, so these numbers cannot be directly compared with those given by the Sharma et al. (2020) paper.

The intrinsic reward during the offline phases on the MuJoCo tasks are:

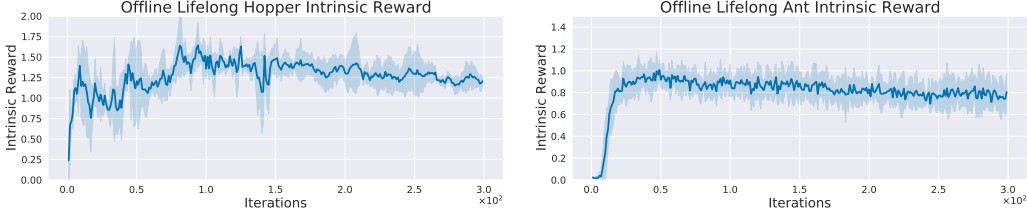

Figure 11: Intrinsic reward during the offline phase of training. Note that these are substantially different than those from episodic training, highlighting the role of exploration in skill discovery.

## C    EPISODIC EVALUATION OF EMPOWERMENT

In this section, we consider LiSP as an episodic learning algorithm (i.e. in the standard skill discovery setting). In particular, to measure performance here we consider how LiSP maximizes *empowerment* (Klyubin et al., 2005; Salge et al., 2014; Gregor et al., 2016), which roughly corresponds to how stable the agent is, or how much "potential" it has. Empowerment is mathematically defined by the mutual information between the current state and future states given actions, and has motivated many skill discovery objectives, including the one we studied in this paper. In fact, this can give benefits to reset-free learning as well: if all the skills maintain high empowerment, then this ensures that the agent only executes stable skills. Although we find we are not able to solely use this type of constraint for safe acting – instead having to rely on long horizons – these types of skills help to reduce the search space for planning. For example, learning good skills to reduce the search space for planning was very beneficial in the 2D Minecraft environment, so this is another experiment yielding insight to this property.

We compare LiSP to DADS (Sharma et al., 2019); note that the main differences will be due to using model-generated rollouts for training the skill policy and the skill-practice distribution. We perform training in the standard episodic Hopper setting (without demos) for skill discovery algorithms, and approximate the empowerment by measuring the average $z$-coordinate of the Hopper, corresponding to the agent's height, as proposed in Zhao et al. (2021). To make this evaluation, we remove the planning component of LiSP, simply running the skill policy as in DADS, and maintain the standard LiSP algorithm otherwise, where both the skill policy and discriminator train exclusively from model rollouts. The changes to our hyperparameters are:

- Every 20 epochs, sample 2000 environment steps

- Generated replay buffer size of 20000

- Generate 20000 model rollouts per epoch

- Take 128 discriminator gradient steps per epoch

- Take 256 skill policy gradient steps per epoch

- Take 32 skill-practice distribution gradient steps per epoch

- We do not use a disagreement penalty

Our results are shown in Figure 12. LiSP acts as a strong model-based skill learning algorithm, achieving a $\approx 5\times$ sample efficiency increase over DADS in this metric. The sample efficiency of LiSP is competitive to model free algorithms on the Hopper environment, demonstrating that it is possible to gain significant learning signal from few samples by utilizing model rollouts, competitive with state-of-the-art algorithms that focus on only learning one task.

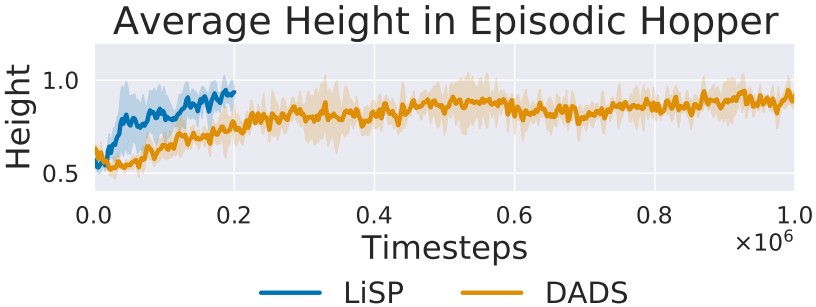

Figure 12: Average height of the hopper for an evaluation episode of length 200 with randomly sampled latents, a proxy for measuring the average empowerment of the learned skills. LiSP shows a considerable improvement in sample efficiency vs DADS in this regard.

## D    OFFLINE LEARNING WITH LIMITED DATA

In this section, we repeat the offline RL experiment from Section 4.4 where we perform offline training of LiSP on a dataset and then use LiSP at test time for multiple tasks without gradient updates, but on a smaller dataset. As described in Appendix E, the original experiment used one million timesteps for training, representing a similar dataset size the to D4RL suite of tasks (Fu et al., 2020). Here, we evaluate on the same dataset but using only the first 200000 timesteps, which represents a small dataset for this task (the replay buffer at the beginning of training), and roughly corresponds to "medium-replay" in D4RL.

The results are shown in Table 2. We include the original results for comparison. Despite the reduced dataset size, LiSP still stays stable although has somewhat worse task performance. Since the size and composition of the dataset was not a focus of our work, we don't include more extensive results, but believe this experiment indicates that LiSP would be promising for varying datasets.

Table 2: Average performance when learning fully offline from a smaller dataset for Lifelong Hopper. The LiSP agent still remains stable in this limited data regime.

| Task | LiSP (200k) | LiSP (1m) | MPC-Long (1m) | MPC-Short (1m) |
|------|-------------|-----------|---------------|----------------|
| Forward Hop – Fast | $0.80 \pm 0.01$ | $\mathbf{0.84 \pm 0.02}$ | $0.27 \pm 0.10$ | $0.27 \pm 0.04$ |
| Forward Hop – Slow | $0.80 \pm 0.01$ | $\mathbf{0.88 \pm 0.05}$ | $0.49 \pm 0.42$ | $0.32 \pm 0.08$ |
| Backward Hop – Slow | $0.79 \pm 0.01$ | $\mathbf{0.81 \pm 0.01}$ | $0.38 \pm 0.16$ | $0.27 \pm 0.06$ |

## E    ENVIRONMENT DETAILS

In this section, we provide additional details about our lifelong experiments and the environment setup, including the modifications to the reward functions, the datasets used for pretraining, and the metrics for performance used.

### E.1    LIFELONG MUJOCO ENVIRONMENTS

Our environment settings were designed such that the optimal behaviors are similar to the optimal behaviors in the Gym version of these environments, and so that the difficulty of learning was similar (although the learning dynamics are different). This is generally accomplished by having some part of the reward for stability, in addition to the standard reward for accomplishing the task, which also emphasizes the health of an agent by considering the reward; standard benchmarks lack this because the signal comes from a termination function. While this extra shaping of the reward gives more signal than normal, these settings are still hard for standard RL algorithms as highlighted above, and even episodic RL without early termination using the standard rewards tends to fail when episodes are long relative to time-to-failure.

In this work, we considered two MuJoCo tasks: Lifelong Hopper and Lifelong Ant. The reward functions we use for these environments are as follows, where $z$ is the height of the agent, $x_{vel}$ is the x-velocity, and $\hat{x}_{vel}$ is a target x-velocity:

- Lifelong Hopper: $-5(z - 1.8)^2 - |x_{vel} - \hat{x}_{vel}| + |\hat{x}_{vel}|$
- Lifelong Ant: $-(z - 0.7)^2 - |x_{vel} - \hat{x}_{vel}| + |\hat{x}_{vel}|$

For the experiments in Section 4.1, we change the target velocity every 1000 timesteps according to the following ordering (chosen arbitrarily):

- Lifelong Hopper: $[0, 1, -1, 2, -1]$
- Lifelong Ant: $[1, -1, 1]$

The dataset we used for both tasks was the replay buffer generated from a SAC agent trained to convergence, which we set as one million timesteps per environment. This is the standard dataset size used in offline RL (Fu et al., 2020), although of higher quality. Despite this, our setting is still

difficult and highly nontrivial due to sink states and gradient instability. We hope future works will continue to explore non-episodic settings with varying datasets.

### E.1.1 PERFORMANCE METRICS

To summarize performance easily in our experiments, we use a performance metric such that it is possible to achieve an optimal performance of 1, and a representative "worst-case" performance is set to 0. This allows for easy comparison between tasks and represents difficulty in harder tasks. The weighting of the performance is similar to the standard reward functions, prioritizing stability with the height term and rewarding competence with a task-specific term, and easily represents the overall gap to a relatively perfect policy/desired behavior with a single number.

For Lifelong Hopper, take the average height $z_{avg}$ of the evaluation duration and the average x velocity $x_{vel}$. Note that it is impossible to make a similar normalization to 1 with returns, since it is impossible to attain the optimal height and velocity from every timestep, but the averages can be optimal. For some target velocity $\hat{x}_{vel}$, the performance we use is given by:

$$1 - 0.8 \cdot \frac{1}{1.3^2}(z_{avg} - 1.3)^2 - 0.2 \cdot \frac{1}{4}|x_{vel} - \hat{x}_{vel}|$$

The height is derived from approximately the height of an optimal SAC agent on the standard Hopper benchmark, representing the behavior we aim to achieve.

For Lifelong Ant, we use a similar metric:

$$1 - 0.5 \cdot \frac{1}{0.5^2}(z_{avg} - 0.7)^2 - 0.5 \cdot \frac{1}{4}|x_{vel} - \hat{x}_{vel}|$$

### E.2 2D GRIDWORLD ENVIRONMENTS AND SKILL VISUALIZATIONS

In this work, we considered two continuous 2D gridworld tasks: volcano and 2D Minecraft. The volcano environment allows us to consider the safety of agents, and Minecraft is a well-known video game setting that is attractive for hierarchical RL due to the nature of tools (Guss et al., 2019); our version is a simple minimal 2D version that is faster to train. We visualize both these environments and skills learned on them below:

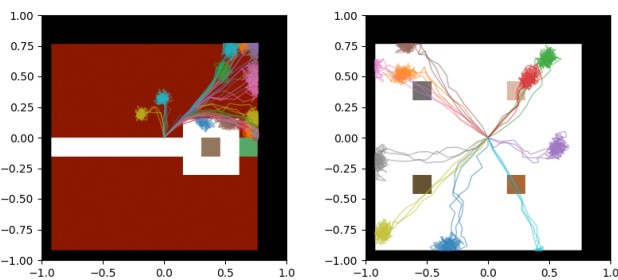

Figure 13: Volcano environment (left) and 2D Minecraft environment (right). Colors denote different skills/latents learned by LiSP. Most skills seek to perform behaviors relevant to the task (reaching the goal, interacting with the tiles), due to being trained on the data distribution of the agent.

### E.2.1 VOLCANO ENVIRONMENT

In the volcano environment, there is lava (shown in dark red), which the agent is penalized for standing in but does not inhibit its movement. The green tile, located on the right side in the picture, denotes the target goal. The reward is given by a negative L2 distance to this goal. The white squares denote free tiles. The primary square of interest is the brown tile, representing a pitfall, where the agent will be trapped if it falls in. The state is the agent's x-y position, the position of the pitfall, as well as the position of the goal. We can see that the learned skills correspond to avoiding the tile, representing how the skills themselves can embed safety, whereas planning in action space would

not achieve the same effect. This behavior follows from an empowerment objective (see Appendix C); in particular, LiSP is capable of offline evaluation of empowerment. We initialize the dataset with 100k demonstrations of a trained SAC agent with noise applied to actions.

### E.2.2 2D MINECRAFT ENVIRONMENT

In the 2D Minecraft environment, the agent maintains an inventory of up to one of each item, and receives a reward whenever obtaining a new item (either through mining or crafting) in increasing magnitudes depending on the hierarchy of the item (how many items were needed to obtain beforehand to obtain the item). The state is the position of the agent, the position of the blocks it can interact with, as well as its inventory, represented by a vector of 0's or 1's to represent ownership of each item. There are four tiles of interest:

- A crafting table (bottom right), where the agent will craft all possible tools using the current materials in its inventory
- A wood block (bottom left), where the agent will obtain a wood for visiting
- A stone block (top left), where the agent will obtain a stone for visiting if it has a wooden pickaxe, consuming the pickaxe
- An iron block (top right), where the agent will obtain an iron for visiting if it has a stone pickaxe, consuming the pickaxe and yielding the maximum reward in the environment

The skill progression is representing by the following ordering:

1. Walk to wood to mine
2. Bring wood to craft table to craft a stick
3. Craft wooden pickaxe with both a wood and a stick
4. Walk to stone to mine using a wooden pickaxe
5. Craft stone pickaxe with both a stone and a stick
6. Walk to iron to mine using a stone pickaxe

There is skill reuse in that lower-level skills must be executed multiple times in order to accomplish higher-level skills, forming a hierarchy of skills in the environment. The dataset consists of 2000 steps of manually provided demonstrations, which is a very low amount of demonstrations.

## F HYPERPARAMETERS

Our code can be found at: https://github.com/kzl/lifelong_rl.

For all algorithms (as applicable) and environments, we use the following hyperparameters, roughly based on common values for the parameters in other works (note we classify the skill-practice distribution as a policy/critic here):

- Discount factor $\gamma$ equal to 0.99
- Replay buffer $\mathcal{D}$ size of $10^6$
- Dynamics model with three hidden layers of size 256 using tanh activations
- Dynamics model ensemble size of 4 and learning rate $10^{-3}$, training every 250 timesteps
- Policy and critics with two hidden layers of size 256 using ReLU activations
- Discriminator with two hidden layers of size 512 using ReLU activations
- Policy, critic, discriminator learning rates of $3 \times 10^{-4}$ training every timestep
- Automatic entropy tuning for SAC
- Batch size of 256 for gradient updates
- MPC population size $S$ set to 400
- MPC planning iterations $P$ set to 10

- MPC number of particles for expectation calculation set to 20
- MPC temperature of 0.01
- MPC noise standard deviation of 1
- For PETS, we use a planning horizon of either 25 or 180, as mentioned
- For DADS, we find it helpful to multiply the intrinsic reward by a factor of 5 for learning (for both DADS and LiSP usage)

For LiSP specifically, our hyperparameters are:

- Planning horizon of 180
- Repeat a skill for three consecutive timesteps for planning (not for environment interaction)
- Replan at every timestep
- Number of rollouts per iteration $M$ set to 400
- Generated replay buffer $\hat{D}$ size set to 5000
- Number of prior samples set to 16 in the denominator of the intrinsic reward
- Number of discriminator updates per iteration set to 4
- Number of policy updates per iteration set to 8
- Number of skill-practice updates per iteration set to 4
- Disagreement threshold $\alpha_{thres}$ set to 0.05 for Hopper, 0.1 for Ant
- Disagreement penalty $\kappa$ set to 30

