# OpenReview forum: "Reset-Free Lifelong Learning with Skill-Space Planning"
_ICLR.cc/2021/Conference — ICLR 2021 Poster_

### Official Review · AnonReviewer2 · 2020-10-28
**Nice Idea**

**Rating:** 6
**Confidence:** 3

**Review:**

Edit: Having gone through the updates and the author's replies, I am increasing my score.


## Summary

* The paper proposes a lifelong reinforcement learning (LRL) approach that (i) learns a model of the env, (ii) uses that model to learn skills, (iii) plan in the abstract space of skills.

## Strengths

* Many design choices make it a good fit for real-life problems:
  *  Skills can be discovered from both offline and online data.
  *  Long horizon planning

* The paper is based on an excellent intuition -- planning in the skill-space. I think the paper describes a reasonable instantiation of the idea. This instantiation is likely to work in practice.

* The paper considers interesting experiments and ablations (though I can't entirely agree with all the results/conclusions -- more on this later). The experiments in the appendix are interesting too.

## Areas to improve

* The paper is somewhat dense to read. In some cases, I had to read the same text three times to make sure I do not misunderstand anything.

* This point is more of a clarification question - Why are the skills sampled from [-1, 1]^dim (page 2 "Skill Discovery")

* It would help if the paper had a more in-depth discussion about how $q_v$ and $p_{\phi}$ are trained. For example, it is not obvious to me if the gradients flow to $p_{\phi}$ (via $z_i$) when training $q_v$.

* It would help if the authors discuss the complexity of the proposed approach, especially the `GetAction` method.

* One significant weakness of the paper is the lack of comparisons with the baselines, which makes it hard to understand the usefulness of this work. For example, consider figure 1 - the reported metric "performance" is defined by the authors, making it difficult to understand how good is a "performance" of say 0.8. Having some other baselines would make the analysis more meaningful. In Figure 4, SAC should be seen as a lower bound for the baselines, and having some other, more reasonable non-stationary RL baselines would be useful. Similarly, in figure 5, it appears that using skills (vs. using actions) is much better because it had a lower variance. But just looking at figure 5, it is difficult to estimate how much lower is the variance (and the mean). Looking at the plot, I can not estimate the mean of the two distributions and how many samples are near the extreme model errors. This can be easily fixed by plotting the l2 error on the x-axis and pdf on the y-axis. One more thing I want to highlight here: The comparison does not seem to be fair because the "skills" are sampled from the trained skill-policy while the "actions" are sampled from a random policy. The model has been trained on the skills sampled from the trained policy, so the result is not surprising.

* One broad concern I have is in understanding the novelty/setup of the paper. My understanding is that the paper claims they improve the performance for reset free lrl (problem setup) using skill space planning (solution setup)? If yes, should they compare against action planning based approaches? I think they try to clarify this in Figure 7, but I did not understand what LOOP agent is

Overall, I think this is exciting and useful work. I have some questions (which I described above), and I might have mis-understood something.  I would be happy to update my score based on the author's answer.

---

> ### Author Response · Authors · 2020-11-19
> **Response to reviewer 2**
>
> Thank you for your review! We appreciate your feedback and our excited that you think our setup is well-grounded and that our experiments are interesting. We have addressed the concerns brought up in your review, and would also be happy to address or implement any further concerns/experiments you suggest.
>
> “The paper is somewhat dense to read.”
>
> We’ve added clarity in some places and hopefully this combined with the other points brought up by other reviewers makes the writing more clear.
>
> “This point is more of a clarification question - Why are the skills sampled from [-1, 1]^dim (page 2 "Skill Discovery")”
>
> Although in principle we do not have to bound the skills, this is the standard convention in RL skill discovery literature and mimics the representation of actions.
>
> “It would help if the paper had a more in-depth discussion about how $q_v$ and $p_{\phi}$ are trained. For example, it is not obvious to me if the gradients flow to $p_{\phi}$ (via $z_i$) when training $q_v$.”
>
> We’ve added some additional clarity here. To be explicit the gradients do not flow through p_phi when training q_v, which we have also added in Section 3.1.
>
> “It would help if the authors discuss the complexity of the proposed approach, especially the GetAction method.”
>
> Our work does not introduce novel components affecting the complexity of GetAction, i.e. it is equivalent to other MPC works except for some additional overhead of running the skill policy. The complexity is linear in the horizon length. We have included this discussion in the paper in Section 3.2.
>
> “One significant weakness of the paper is the lack of comparisons with the baselines, which makes it hard to understand the usefulness of this work...” (note: is Figure 1 supposed to say Figure 3 here?)
>
> In general, we find the key difficulty in these settings stems from the lack of resets, so we do not explicitly include non-stationary baselines. We’ve added MPC (the most natural multi-task baseline) and MOReL (a model-based offline algorithm as discussed in the paper as Kidambi et al. 2020) as baselines to Figure 3 and 4, as well as MPC to Figure 8. We added a small discussion of these baselines in Section 5, in particular finding that LiSP outperforms them.
>
> “Similarly, in figure 5, it appears that using skills (vs. using actions) is much better because it had a lower variance. But just looking at figure 5, it is difficult to estimate how much lower is the variance (and the mean). Looking at the plot, I can not estimate the mean of the two distributions and how many samples are near the extreme model errors. This can be easily fixed by plotting the l2 error on the x-axis and pdf on the y-axis.”
>
> Plotting the pdfs would be fairly cluttered and look very similar; in fact around 97.4% of the policy actions and 90.4% of the random actions have errors less than 0.005, so almost all of the pdf would be in a small portion of a graph (currently represented by the boxplot rectangle). The mean and variance are quite similar but the tails are the important aspect of the figure. We think that the current visualization highlights the long tail of the action errors; if this tail is big, even if most of the actions have small errors, it is possible (and likely) for MPC to critically overestimate the returns on this tail.
>
> “One more thing I want to highlight here: The comparison does not seem to be fair because the "skills" are sampled from the trained skill-policy while the "actions" are sampled from a random policy. The model has been trained on the skills sampled from the trained policy, so the result is not surprising.”
>
> While the skills are sampled from a trained policy, it is nontrivial in general to represent such a constraint -- and learning skills in this fashion does implement this. The L2 error of the observation is not something we would have access to during live evaluation, so, while not surprising, it is important that we try the predictive capacity to the L2 error. Furthermore, our own experiments attempting to apply a disagreement penalty directly to the action space also led to poor performance for MPC, which would otherwise a natural choice to represent the constraint.
>
> “One broad concern I have is in understanding the novelty/setup of the paper. My understanding is that the paper claims they improve the performance for reset free lrl (problem setup) using skill space planning (solution setup)? If yes, should they compare against action planning based approaches? I think they try to clarify this in Figure 7, but I did not understand what LOOP agent is”
>
> Yes, this is the key contribution of our paper and is the main baseline we compare against! We apologize for the typo in Figure 7 where we meant to write MPC instead of LOOP, which was very confusing. In particular, we compare against action-space MPC in Figures 5, 6, and 7, as well as in Table 1. During the rebuttal phase,we’ve now added MPC to Figures 3, 4, and 8 as well.

---

### Official Review · AnonReviewer1 · 2020-10-28
**Important topic, promising results.**

**Rating:** 6
**Confidence:** 3

**Review:**

### Summary
The authors propose LiSP, a model-based planning method that performs model-predictive control using learned skills rather than actions. The skills are learned using DADS, with a modified reward function that additionally encourages all skills to stay within the support of training data to avoid sink states. The experiment results show stable learning progress on reset-free and ever-changing targets, compared to other baselines.

Overall, the specific topic of lifelong setting (i.e., reset-free learning) studied in the paper is pretty relevant, and the main results of the proposed method are promising.

### Strength
- The problem is significant and pretty relevant to the field.
- LiSP seems to work well on the reset-free environments considered in the paper.

### Weakness
- The scope of non-stationarity investigated here is pretty narrow (only focus on non-stationarity of tasks, as mentioned in the appendix). It is very likely that the proposed framework will fail in other cases (e.g., change of terrain, etc).
- Analysis of the newly introduced hyperparameters $\kappa$ and $alpha$ (for the adjusted reward) is missing. How sensitive is LiSP to these parameters?
- Figure 8: what do we want to test for this "minimizing intrinsic reward" case? What's the intuition about it?
- While the authors have shown some pretty interesting observations/analysis, the writing is a little bit messy. So what exactly helps the agent in the reset-free setting / avoid the sink states? Is it because of the longer-step planning (as shown in Figure 7)? Or is it simply because the intrinsic rewards drive the skills to visit different states? Also, how can we tell from Figure 7 and Figure 1 that MBPO fails because of the instability of gradients? We only have numerical results on the environment returns here.

### Additional Comments
- I am willing to change my score based on the authors' response to the problems raised in mine and other people's reviews.

=============== Edit ===============

After reading the authors' response to me and other reviewers, I think my concerns are sufficiently addressed.
Therefore, I update my rating from 5 to 6.

There is still room for improvement in terms of the writing: as raised by the other reviewers, the text is a little bit dense to read.
It would be great if the authors can further refine and brush-up the flow of the paper to make it more accessible.

---

> ### Author Response · Authors · 2020-11-19
> **Response to reviewer 1**
>
> Thank you for your review! We are excited that you think our results are promising and the setting is relevant. We have addressed most comments here, but we welcome further comments or suggestions which we will address through the remainder of the rebuttal phase:
>
> “The scope of non-stationarity investigated here is pretty narrow (only focus on non-stationarity of tasks, as mentioned in the appendix). It is very likely that the proposed framework will fail in other cases (e.g., change of terrain, etc).”
>
> While the scope of non-stationarity is fairly narrow, there is a significant added difficulty challenge from the lack of resets and presence of sink states, which has previously been very underexplored. Our work focuses principally on this added difficulty, and it is possible that later algorithmic improvements or combination with existing continual learning systems could be used for more complex systems if LiSP does not perform well in them.
>
> “Analysis of the newly introduced hyperparameters $\kappa$ and $alpha$ (for the adjusted reward) is missing. How sensitive is LiSP to these parameters?”
>
> We would like to note these hyperparameters were introduced by the Kidambi et al. 2020 work referenced in the paper, so they are not newly introduced. In our experience, we don’t find LiSP to be very sensitive to these, but we will follow up with a more detailed analysis.
>
> “Figure 8: what do we want to test for this "minimizing intrinsic reward" case? What's the intuition about it?”
>
> This is simply a sanity check that we should maximize the intrinsic reward (instead of, say, any trivial distribution working).
>
> “So what exactly helps the agent in the reset-free setting / avoid the sink states? Is it because of the longer-step planning (as shown in Figure 7)? Or is it simply because the intrinsic rewards drive the skills to visit different states?”
>
> We added a line in Section 4.1 that the long-horizon component is crucial, which is also discussed in Section 4.4. In order to enable long-horizon planning, we utilize planning with skills, as highlighted in Figures 5 and 6. We have expanded our discussion of the experimental results in line with more baselines that help to illustrate the main insights.
>
> “Also, how can we tell from Figure 7 and Figure 1 that MBPO fails because of the instability of gradients? We only have numerical results on the environment returns here.”
>
> In Figure 1, the curves are distinguished from the red line because they additionally take gradient steps, whereas the red line represents freezing the weights of the policy network after training; this is the only difference. This shows that it is the gradient updates that makes the agents unstable. This setup is repeated in Figure 7.

---

> > ### Comment · AnonReviewer1 · 2020-11-20
> > **Thanks for addressing my questions**
> >
> > Thanks for all your efforts in addressing my concerns. I have changed my rating correspondingly.
> >
> > Just one additional comment (not really related to your method): it will be easier for reviewers to locate where you change or added new texts if you highlight them in color (red, blue, etc).

---

> > > ### Author Response · Authors · 2020-11-25
> > > **Hyperparameter sensitivity experiment**
> > >
> > > We've now added an experiment for hyperparameter sensitivity in Appendix G; our findings is that it is not sensitive to the range we tested.

---

### Official Review · AnonReviewer3 · 2020-10-28
**This work presents the LiSP architecture for model-based learning with non-stationary rewards, and applies it to the problem of lifelong-learning in environments without state resets.  Empirical results demonstrate that LiSP, which uses a skill-space predictive model and planning algorithm, is more robust to the lack of state resets than either action-space model-predictive control or model-free RL.**

**Rating:** 7
**Confidence:** 3

**Review:**

SUMMARY OF CONTRIBUTION:

This work presents the LiSP architecture for model-based lifelong learning and applies it to the problem of learning in continuing tasks with non-stationary rewards and no state resets.  LiSP builds on the DADS algorithm, which learns predictive dynamics model of an RL environment that depends on a continuous space of high-level skills, rather than primitive actions, in an unsupervised fashion that does not require an external reward function.  Unlike DADS, LiSP appears to learn an ensemble of primitive models, which is then used to generate simulated transitions from which the DADS algorithm learns its high-level model.  The goal of this work is to utilize the high-level predictive model learned via DADS to address two issues, reset free learning, and learning with non-stationary reward functions.  In this work, DADS is used to learn a dynamics model during a pretraining phase, and this model is then used as a component of a model-predictive control algorithm to select actions for the non-stationary, reset-free task.  The use of a model-based planning allows the agent to act optimally with respect to a non-stationary reward function (the agent has access to the full reward function at each time step).  The fact that the agent does not need to learn its policy from scratch potentially makes this approach far more robust to the permanent failure states.  The experimental results show that LiSP is effective in reset-free, non-stationary reward versions of the MuJoCo Hopper and Ant tasks, as well as a 2D Minecraft--like task.

AREAS OF CONCERN:

One of the main issues with the paper as it is currently written is that the description of the LiSP algorithm is not detailed enough to understand exactly what the proposed method does, and how it differs from the existing DADS algorithm.  Specific questions that need to be answered are:
  1) How is the primitive dynamics model f_phi trained?
    a) How much data is used to train this model?
    b) What exploration policy is used to update this model?
    c) How accurate are the primitive models?
  2) The description of LiSP alternates between describing a single model and an ensemble of models, how is this ensemble trained, and how are transitions sampled from it?
  3) What loss is used to train the practice distribution p_psi?

The main concern with the current set of experiments is that they do not seem to isolate the two different sources of failure in the lifelong learning setting, that is, the lack of state resets and the non-stationarity of the reward function.  In Figure 1, it seems likely that MBPO fails due to the existence of a sink state, but it is possible that SAC is also failing due to a change in the reward function.  It seems unfair to compare SAC to MPC in the non-stationary reward setting, as a model-free algorithm will inevitably lose out in this setting to an MPC algorithm which can adapt instantly to a new reward.  It would be more informative to compare LiSP, MBPO and SAC in terms of the stability of their learned policies alone.  If the source of the instability of SAC is the combination of non-stationary rewards and sink-states, then ablation where only one or the other of these conditions is present would be useful to highlight this.

It is also unclear whether continuing to update the model and skill policy online is necessary.  Figure 7 suggests that not updating the skill policy actually improves stability, so it seems likely that freezing the predictive model would be beneficial as well, or would at the very least not lead to a meaningful loss of performance.

The results shown in Figure 8 are promising, but their significance is difficult to interpret.  It might be helpful to provide a more detailed discussion of the learned practice distribution, and why it leads to such a substantial improvement over the uniform distribution used in the original DADS algorithm.  It was originally suggested (Section 3.1) that the learned practice distribution is needed due to the use of a primitive model to generate synthetic training data.  It would be helpful to include an ablation which shows the differences in performance between a uniform and learned skill distribution when a primitive model is not used, and the DADS skill model is trained on raw transitions.

There are a few other minor questions that need to be addressed:
  1) What does the LOOP agent mentioned in the caption for Figure 7 refer to?
  2) In Figure 5, it appears that what we are evaluating is the error of the primitive (action-conditional) predictive model, rather than the high-level model, but this is not explicitly state.
  3) It isn't entirely clear what the sink state for the Ant task looks like, as unlike Hopper, it would seem that even if the Ant falls down it should be able to push itself back up (The fact that we don't observe any runs failing for Ant in Figure 3 suggests that Ant has no sink state).
  4) In what experiments is pretraining being used?  While it appears that the policies used in the Hopper environment are pretrained, it is not clear that pretraining is used in the Minecraft environment (Figure 8).

CONCLUSION:

The core idea of this work is sound, and addresses major limitation of reinforcement learning in real-world settings, namely, the need for state resets.  While the algorithm presented is incremental, results suggesting that skill-space model-predictive control may be robust in reset-free learning where primitive MPC appears to fail.  While the experimental results presented here are somewhat limited, it is likely that other researchers could use them as the basis for more detailed investigation of the problem of reset-free learning.  At the very least, however, the presentation of the algorithm and the experimental results needs significant improvement, and my score reflects the current state of the paper, but could increase if the presentation issues above were addressed.

---

> ### Author Response · Authors · 2020-11-19
> **Response to reviewer 3**
>
> Thank you for your review! We are excited that you see the need for state resets as a crucial weakness of reinforcement learning. Most of the concerns brought up seem to be from presentation issues, so we have edited the paper based on the feedback and hope to use the remainder of the rebuttal phase to address any additional concerns.
>
> “One of the main issues with the paper as it is currently written is that the description of the LiSP algorithm is not detailed enough...”
>
> LiSP is designed as an algorithm for reset-free, nonstationary learning via its capability to learn a policy without minimal environment interactions and use of long-horizon planning to ensure safety; we only use DADS to generate intrinsic rewards. We added some discussion of the key components of LiSP at the beginning of Section 3.
>
> “How is the primitive dynamics model f_phi trained? a) How much data is used to train this model? b) What exploration policy is used to update this model? c) How accurate are the primitive models?”
>
> Each model in the ensemble is trained independently using a standard loss to maximize the log-probability of the transitions; we added a clarification here that specifically it is f_phi(s_t+1 | s_t, a_t). (a) The model is trained on the replay buffer, similar to other MBRL works. The starting datasets consists of a few thousand timesteps for the Minecraft and Volcano environments, and 1 million for the MuJoCo tasks, which we further describe in Appendix C. (b) New transitions are collected via Algorithm 3, “GetAction”. (c) In Section 4.2, we show the L2 error in the primitive models for the Hopper environment: for actions derived from skills, almost all errors are bounded by about 0.04, whereas for arbitrary actions the errors are roughly bounded by 0.09.
>
> “The description of LiSP alternates ... how is this ensemble trained, and how are transitions sampled from it?”
>
> We added some discussion in Section 2; each model is trained independently. For sampling, MPC uses trajectory sampling. For rollout generation, we pick one member of the ensemble uniformly at random (we clarify this point in Section 3.1, in the paragraph starting “To actually learn the policy…”).
>
> “What loss is used to train the practice distribution p_psi?”
>
> From Section 3.1, we optimize p_psi jointly with the policy pi_theta by maximizing the intrinsic reward shown in Equation 3 using SAC.
>
> “The main concern with the current set of experiments is that they do not seem to isolate the two different sources of failure in the lifelong learning setting...”
>
> This is very important! In particular, in Figures 1 and 7 (representing the same setting), we are isolating the challenges due to lack of state resets specifically -- there is no change in reward function or dynamics. Here, we argue that algorithms which do not perform planning are inherently unstable without resets. Even when the agent is already fully trained and there is no change in the reward, SAC and MBPO do not remain stable with gradient updates (Figure 1),  in stark contrast to the stable behavior of MPC with a terminal value function (Figure 7).
>
> “It is also unclear whether continuing to update the model and skill policy online is necessary...”
>
> In non-stationary settings, we believe it is useful to have algorithms which maintain this property as they are better able to adapt to the current environment, as well as to continue learning in general. Part of the excitement of this setting is that it captures the following notion: assume an agent is trained thoroughly in a training environment/simulator, but which cannot capture all possible scenarios at deployment time. Then, upon encountering new settings and/or user specifications, it can adapt during deployment phase, allowing for better generalization.
>
> “The results shown in Figure 8 are promising, but their significance is difficult to interpret. It might be helpful to provide a more detailed discussion of the learned practice distribution, and why it leads to such a substantial improvement over the uniform distribution used in the original DADS algorithm.”
>
> Note that in Figure 9 we do not use a primitive dynamics model, showcasing how uniform resampling of skills can inhibit skill learning. Intuitively, during a long rollout, you are able to generate many correlated transitions from the same skill, and associate states visited along that transition with that skill. However, during a short or one-step rollout, there is no notion of associating states visited with skills, if the skills are sampled uniformly. A skill-practice distribution assigning skills to states can help reinstate this notion. We added this intuition to the paper in Section 4.3.
>
> Minor issues:
> (1) This was a typo, we meant to say MPC
> (2) We explicitly added f_phi here now
> (3) These sink states correspond to the Ant flipping over; LiSP tends to be conservative so that this does not occur
> (4) Pretraining is used in all envs except Figures 4 and 8, which we now specify

---

> > ### Comment · AnonReviewer3 · 2020-11-23
> > **Response to rebuttals**
> >
> > I believe the authors have addressed most of my concerns, and have updated my scores to reflect this.
> >
> > A couple of remaining points:
> >
> > Regarding question 1), I'm still not entirely sure I understand the answer to part b).  I was specifically interested in how the data that initially populates the replay buffer is generated, and what policy is followed to generate that data.
> >
> > It would be useful to provide separate results for lifelong-learning and reset-free learning for all environments, including the Ant and Minecraft environments.

---

> > > ### Author Response · Authors · 2020-11-25
> > > **Consideration of datasets**
> > >
> > > Thank you again for your response! Your feedback has been very helpful for improving the paper.
> > >
> > > As for question 1, we included some details here in Appendix E, and generally it's some variation of generated by a SAC policy (or manual demos for 2D Minecraft); for the MuJoCo tasks it's 1 million timesteps (we made the datasets more clear in our new revision). We didn't focus on this in our work which is why we emphasized other investigations, but we also now include a new section Appendix D "Offline Learning with Limited Data" training on the first 200 thousand timesteps instead, and find LiSP is also stable in this regime as well. We also note that in a new experiment (Appendix C), we perform episodic training, which also naturally has access to less data and LiSP also performs well.
> > >
> > > Due to training time it will not be as easy to provide more detailed experiments for the Ant and Minecraft environments, especially within this discussion phase, but we will try to think of an insightful experiment here as well to add.

---

### Official Review · AnonReviewer4 · 2020-10-28
**like the topic, have doubts about problem formulation**

**Rating:** 5
**Confidence:** 3

**Review:**

This paper presents a lifelong reinforcement learning framework in a non-stationary environment with non-episodic interactions. The proposed approach is to 1) learn "skills" - a world model - to maximize the intrinsic rewards using both online and offline data, and to 2) make best plans based on the learned world model. This approach is evaluated with Hopper and Ant tasks.

Overall, I believe this paper is marginally below the acceptance borderline. I like the keywords - intrinsic rewards, catastrophic forgetting, non-stationary environment, model-based reinforcement learning. But I doubt the usefulness of the framework. The evaluation based on performing two tasks and insufficient comparison with other algorithms is another minus.

Pros:

+ Lifelong learning is a machine learning area with a long history and the keywords such as intrinsic rewards, catastrophic ..., non-stationary environment, model-based reinforcement learning all shows that this paper is in the right direction.
+ The formulation of "skill" - world model - as a latent variable is interesting
+ Model-based reinforcement learning is an underexplored area by my assessment.

Cons:

- The evaluation is too simple in comparison with the goal of lifelong learning. I would like to see how the agent performs for multiple tasks.
- I would like to see how agents improve performance in new tasks, and at the same time, retain the performance of old tasks.
- I have doubts that the skill formulation with some simple prior $p(z)$ is sufficient for complex tasks.
- The formulation is less ambitious than what I would expect an accepted paper to be: this paper seems to learn a some-what fixed environment and just solve the tough model-based reinforcement planning problem with mpc. There are many potential issues with mpc in a complex world and I see no performance guarantee.
- I hope to see how previous learned task knowledge could contribute to future learnings. But the proposed framework doesn't seem to accomondate that.

---

> ### Author Response · Authors · 2020-11-19
> **Response to reviewer 4**
>
> Thank you for your review! We are also excited by the promise of combining skills with model-based RL and lifelong learning. At a high level, our primary contribution is progress towards solving reset-free settings with sink states, which is a novel contribution as past reset-free/non-episodic works have either assumed: access to the ground-truth dynamics model [1], availability of some resets [2], or only evaluate in environments without sink states [3]. This setting is interesting because the world can change or is in some sense unknown, which is why we believe continual learning has a synergistic relationship with reset-free learning. As a result there is a need to further explore and/or adapt to the world in a more risk-critical setting than the original training phase, i.e. when we no longer assume access to resets. Based on this, here are some finer points:
>
> “The evaluation is too simple in comparison with the goal of lifelong learning. I would like to see how the agent performs for multiple tasks.”
>
> Past work in continual RL has focused on catastrophic forgetting or fast adaptation; our settings focus on safe adaptation when resets are not accessible, making it fundamentally harder as it is an added challenge. In particular, the number of tasks in one evaluation in the former is limited only by the capacity of the agent to remember different tasks, whereas in our setting it is limited by the capability of the agent to stay safe. Also, we have now benchmarked several other algorithms in order to better demonstrate the capabilities of LiSP and various insights into the lifelong setting. Combined with our previous experiments focused on hierarchical skill learning in 2D Minecraft and safety in the Volcano environment, we have shown the benefits of LiSP on a wide variety of nonstationary, reset-free environments. That being said, we think this is a good point and the additional experiments are valuable! Please let us know if there are any specific additional experiments that would be useful and can be reasonably executed in the remaining time frame.
>
> “I would like to see how agents improve performance in new tasks, and at the same time, retain the performance of old tasks.” / “I hope to see how previous learned task knowledge could contribute to future learnings. But the proposed framework doesn't seem to accommodate that.”
>
> We assume that the agent is capable of optimizing against an arbitrary specified task, which includes previous tasks, which is why we don’t test previous tasks explicitly. Note that agents do perform well in new tasks because they have prior experience and adapt quickly via planning.
>
> “I have doubts that the skill formulation with some simple prior $p(z)$ is sufficient for complex tasks.”
>
> It is pretty difficult to know how well this practice-distribution would scale (also, note it is p(z|s), not p(z), which is a subtle but important distinction). However, in our new benchmark of MPC in the 2D Minecraft environment, we see that it helps planning perform much better than action-space MPC in this more complicated hierarchical task, so we think there is promise that it should continue to perform well.
>
> “The formulation is less ambitious than what I would expect an accepted paper to be: this paper seems to learn a some-what fixed environment and just solve the tough model-based reinforcement planning problem with mpc. There are many potential issues with mpc in a complex world and I see no performance guarantee.”
>
> MPC certainly has its limitations. However, in this work we push the boundary of what MPC can solve by performing planning using skills, which increases the flexibility of MPC, allowing for model errors both to be constrained and to be alleviated via the generalizability of a model-free policy, as indicated by the Hopper and Ant experiments. We also show that planning in the skill space greatly improves the exploration capacity of MPC in the 2D Minecraft experiment. We expect more works, such as other literature exploring combining model-based and model-free architectures, to continue improving the capabilities of MPC.
>
> [1] Lu et al. 2019. “Adaptive Online Planning for Continual Lifelong Learning.”
>
> [2] Eysenbach et al. 2017. “Leave No Trace: Learning to Reset for Safe and Autonomous Reinforcement Learning.”
>
> [3] Zhu et al. 2020. “The Ingredients of Real-World Robotic Reinforcement Learning.”

---

### Author Response · Authors · 2020-11-25
**Summary of revisions**

Again, we would like to thank all of the reviewers for their insightful reviews, which we have used to significantly improve our paper! Our work focused on a novel reset-free setting which we think will be important for the real-world deployment of RL agents and we are excited that reviewers are interested in it.

Our revisions are summarized as follows:

**First revision (11/19; comments titled as "Response to reviewer X")**
- We have added more baselines to experiments (as suggested by R1, R2), notably MPC, MOReL, and SAC; MPC is most important for isolating the benefits of skills for planning, and they help to understand quantitative performance
- Improvements to writing (see below)

**Second revision (11/24)**
- New experiment treating LiSP as an episodic RL algorithm and approximating the empowerment of the agent, showing high sample efficiency in skill learning (5x improvement to DADS) and providing more insight into how the skills constrain the search space (Appendix C "Episodic Evaluation of Empowerment"), which further motivates the benefits of planning with skills (related to comment by R1)
- New experiment where LiSP is trained offline with a small dataset (related to comment by R3), showing LiSP can still avoid failures via planning though with somewhat lower performance (Appendix D "Offline Learning with Limited Data")
- New experiment showing hyperparameter sensitivity ($\alpha_{thres}, \kappa$) of LiSP for offline lifelong hopper, finding it is not very sensitive (brought up by R1) (Appendix G "Hyperparameter Sensitivity")
- Improvements to writing (see below)

**Improvements to clarity of writing (both revisions)**
- Clearer description of model training and usage in Sections 2 and 3.1 (R2, R3)
- Added descriptions of networks learned/algorithmic components in Section 3 (R2, R3)
- Added paragraph in Section 4 to describe new baselines, and complementary analysis on them per experiment
- More descriptions and intuitions of the skill-practice distribution (R1, R3)
- Clearer descriptions of datasets used in Appendix E "Environment Details" (R3)
- General improvements to writing clarity and points brought up by reviewers

---

### Decision · Program_Chairs · 2021-01-07
**Final Decision**

**Decision:**

Accept (Poster)

**Comment:**

The reviewers were excited by this work, which focuses on lifelong RL in non-stationary, non-episodic environments.  They found the approach compelling with exciting results on the tested domains.  However, even the more positive reviewers were concerned with the somewhat narrow scope of evaluation, which makes the paper somewhat less ambitious.

In response to the reviews, the authors added extra experiments, clarifying text, and requested details that provide more depth and insight to the paper.  Still, the approach and paper is somewhat narrowly focused, but it does yield insights that should be useful for future works that solve this problem in a more general manner.